# LanczosNet: Multi-Scale Deep Graph Convolutional Networks

**Renjie Liao**[1,2,3]**, Zhizhen Zhao**[4]**, Raquel Urtasun**[1,2,3]**, Richard S. Zemel**[1,3,5]

University of Toronto[1], Uber ATG Toronto[2], Vector Institute[3],
University of Illinois at Urbana-Champaign[4], Canadian Institute for Advanced Research[5]
{rjliao, urtasun, zemel}@cs.toronto.edu, zhizhenz@illinois.edu

## Abstract

We propose the Lanczos network (LanczosNet), which uses the Lanczos algorithm to construct low rank approximations of the graph Laplacian for graph convolution. Relying on the tridiagonal decomposition of the Lanczos algorithm, we not only efficiently exploit multi-scale information via fast approximated computation of matrix power but also design learnable spectral filters. Being fully differentiable, LanczosNet facilitates both graph kernel learning as well as learning node embeddings. We show the connection between our LanczosNet and graph based manifold learning methods, especially the diffusion maps. We benchmark our model against several recent deep graph networks on citation networks and QM8 quantum chemistry dataset. Experimental results show that our model achieves the state-of-the-art performance in most tasks.

## 1 Introduction

Graph-structured data is ubiquitous in real world applications, social networks, gene expression regulatory networks, protein-protein interactions, and many other physical systems. How to model such data using machine learning, especially deep learning, has become a central research question [1]. For supervised and semi-supervised tasks such as graph or node classification and regression, learning based models can be roughly categorized into two classes, formulated either in terms of graph convolutions [2] or recurrent neural networks [3].

Methods based on recurrent neural networks (RNN), especially graph neural networks (GNN) [3], repeatedly unroll a message passing process over the graph by exchanging information between the nodes. In theory, a GNN can have as large a model capacity as its convolutional counterpart. However, due to the instability of RNN dynamics and difficulty of optimization, GNN and its variants are generally slower and harder to train.

In this paper we focus on graph convolution based methods. Built on top of the graph signal processing (GSP) approaches [4], these methods extend convolution operators to graphs by leveraging spectral graph theory, graph wavelet theory, etc. Graph convolutions can be stacked and combined with nonlinear activation functions to build deep models, just as in regular convolutional neural networks (CNN). They often have large model capacity and achieve promising results. Also, graph convolution can be easily implemented with modern scientific computing libraries.

There are two main issues with current graph convolution approaches. First, it is not clear how to efficiently leverage multi-scale information except by directly stacking multiple layers. Having an effective multi-scale scheme is key for enabling the model to be invariant to scale changes, and to capture many intrinsic regularities [5, 6]. Graph coarsening methods have been proposed to form a hierarchy of multi-scale graphs [7], but this coarsening process is fixed during both inference and learning which may cause some bias. Alternatively, the graph signal can be multiplied by the exponentiated graph Laplacian, where the exponent indicates the scale of the diffusion process on the graph [8]. Unfortunately, the computation and memory cost increases linearly with the exponent, which prohibits the exploitation of long scale diffusion in practice. Other fast methods for computing matrix power such as exponentiating by squaring are very memory intensive, even for moderately large graphs. Second, spectral filters within current graph convolution based models

are mostly fixed. In the context of image processing, using a Gaussian kernel along with a spectral filter $f(\lambda) = 2\lambda - \lambda^2$ corresponds to running forward the heat equation (blurring) followed by running it backwards (sharpening) [9]. Multi-scale kernels introduced in [10] extends the idea of forward-backward diffusion process and can be represented as polynomials of matrices related to a Gaussian kernel. Learning the spectral filters is thus beneficial since it learns the stochastic processes on the graph which produce useful representations for particular tasks. However, how to learn spectral filters which have large model capacity is largely underexplored.

In this paper, we propose the Lanczos network (LanczosNet) to overcome the aforementioned issues. First, based on the tridiagonal decomposition implied by the Lanczos algorithm, our model exploits the low rank approximation of the graph Laplacian. This approximation facilitates efficient computation of matrix powers thus gathering multi-scale information easily. Second, we design learnable spectral filters based on the approximation which effectively increase model capacity. In scenarios where one wants to learn the graph kernel and/or node embeddings, we propose another variant, *i.e.*, adaptive Lanczos network (AdaLanczosNet), which back-propagates through the Lanczos algorithm. We show that our proposed model is closely related to graph based manifold learning approaches such as diffusion maps which could potentially inspire more work from the intersection between deep graph networks and manifold learning. We benchmark against 9 recent deep graph networks, including both convolutional and RNN based methods, on citation networks and a quantum chemistry graph regression problem, and achieve state-of-the-art results in most tasks.

## 2 BACKGROUND

In this section, we introduce some background material. A graph $\mathcal{G}$ with $N$ nodes is denoted as $\mathcal{G} = (\mathcal{V}, \mathcal{E}, A)$, where $A \in \mathbb{R}^{N \times N}$ is an adjacency matrix which could either be binary or real valued. $X \in \mathbb{R}^{N \times F}$ is the compact representation of node features (or graph signal in the GSP literature). For any node $v \in \mathcal{V}$, we denote its feature as a row vector $X_{v,:} \in \mathbb{R}^{1 \times F}$. We use $X_{:,i}$ to denote the $i$-th column of $X$.

**Graph Fourier Transform**  Given input node features $X$, we now discuss how to perform a graph convolution. Based on the adjacency matrix $A$, we compute the graph Laplacian $L$ which can be defined in different ways: (1) $L = D - A$; (2) $L = I - D^{-1}A$; (3) $L = I - D^{-\frac{1}{2}}AD^{-\frac{1}{2}}$, where $D$ is a diagonal degree matrix and $D_{i,i} = \sum_{j=1}^{N} A_{i,j}$. The definition (3) is often used in the GSP literature due to the fact that it is real symmetric, positive semi-definite (PSD) and has eigenvalues lying in $[0, 2]$. In certain applications [11], it was found that adding self-loops, i.e., changing $A$ to $A + I$, and using the affinity matrix $S = D^{-\frac{1}{2}}AD^{-\frac{1}{2}}$ instead of $L$ gives better results. Since $S$ is real symmetric, based on spectral decomposition, we have $S = U\Lambda U^\top$ where $U$ is an orthogonal matrix and its column vectors are the eigenvectors of $S$. The diagonal matrix $\Lambda$ contains the sorted eigenvalues where $\Lambda_{i,i} = \lambda_i$ and $1 \geq \lambda_1 \geq \cdots \geq \lambda_N \geq -1$. Based on the eigenbasis, we can define the *graph Fourier transform* $Y = U^\top X$ and its inverse transform $\hat{X} = UY$ following [12]. Note that $L = I - D^{-\frac{1}{2}}AD^{-\frac{1}{2}}$ shares the same eigenvectors with $S = D^{-\frac{1}{2}}AD^{-\frac{1}{2}}$ and the eigenvalues of $L$ are $\mu_i = 1 - \lambda_i$. Therefore, $L$ and $S$ share the same *graph Fourier transform* which justifies the usages of $S$. Different forms of filters can be further constructed in the spectral domain.

**Localized Polynomial Filter**  A $\tau$-localized polynomial filter is typically adopted in GSP literature [12], $g_w(\Lambda) = \sum_{t=0}^{\tau-1} w_t \Lambda^t$, where $\boldsymbol{w} = [w_0, w_1, \ldots, w_{\tau-1}] \in \mathbb{R}^{\tau \times 1}$ is the filter coefficient, i.e., learnable parameter. The filter is $\tau$-localized in the sense that the filtering leverages information from nodes which are at most $\tau$-hops away. One prominent example of this class is the Chebyshev polynomial filter [7]. Here the graph Laplacian is modified to $\tilde{L} = 2L/\lambda_{\max} - I$ such that its eigenvalues fall into $[-1, 1]$. Then the Chebyshev polynomial recursion is applied: $\tilde{X}(t) = 2\tilde{L}\tilde{X}(t - 1) - \tilde{X}(t - 2)$ where $\tilde{X}(0) = X$ and $\tilde{X}(1) = \tilde{L}X$. For a pair of input and output channels $(i, j)$, the final filtering becomes, $y_{i,j} = [\tilde{X}(0)_{:,i}, \ldots, \tilde{X}(\tau - 1)_{:,i}]\boldsymbol{w}_{i,j}$, where $[\cdot]$ means concatenation along columns and $\boldsymbol{w}_{i,j} \in \mathbb{R}^{\tau \times 1}$. Chebyshev polynomials provide two benefits: they form an orthogonal basis of $L^2([-1, 1], dy/\sqrt{1 - y^2})$ and one avoids the spectral decomposition of $\tilde{L}$ in the filtering. However, the functional form of the spectral filter is not learnable, and cannot adapt to the data.

In this paper, instead of using the modified graph Laplacian $\tilde{L}$, we use the aforementioned $S$. Therefore, we can write the localized polynomial filtering in a more general form as,

$$Y = \sum_{t=0}^{\tau-1} g_t(S, \dots, S^t, X)W_t, \tag{1}$$

where $g_t$ is a function that takes node features $X$ and powers of the affinity matrices up to the $t$-th order as input and outputs a $N \times F$ matrix. $W_t \in \mathbb{R}^{F \times O}$ is the corresponding filter coefficient and $Y \in \mathbb{R}^{N \times O}$ is the output. One can easily verify that in the Chebyshev polynomial filter, any $i$-th column of the corresponding $g_t(X, S, \dots, S^t)$ lies in the *Krylov subspace* $\mathcal{K}_{t+1}(S, X_{:,i}) \equiv \text{span}\{X_{:,i}, SX_{:,i}, \dots, S^t X_{:,i}\}$. This naturally motivates the usage of Krylov subspace methods, like the Lanczos algorithm [13], since it provides an orthonormal basis for the above Krylov subspace, thus making the filter coefficients compact.

## 3 LANCZOS NETWORKS

In this section, we first introduce the Lanczos algorithm which approximates the affinity matrix $S$. We present our first model, called Lanczos network (LanczosNet), in which we execute the Lanczos algorithm once per graph and fix the basis throughout inference and learning. Then we introduce the adaptive Lanczos network (AdaLanczosNet) in which we learn the graph kernel and/or node embedding by back-propagating through the Lanczos algorithm.

---

**Algorithm 1** : Lanczos Algorithm

1: **Input:** $S, x, K, \epsilon$
2: **Initialization:** $\beta_0 = 0$, $q_0 = 0$, and $q_1 = x/\|x\|$
3: **For** $j = 1, 2, \dots, K$:
4:     $z = Sq_j$
5:     $\gamma_j = q_j^\top z$
6:     $z = z - \gamma_j q_j - \beta_{j-1} q_{j-1}$
7:     $\beta_j = \|z\|_2$
8:     **If** $\beta_j < \epsilon$, quit
9:     $q_{j+1} = z/\beta_j$
10:
11: $Q = [q_1, q_2, \cdots, q_K]$
12: Construct $T$ following Eq. (2)
13: Eigen decomposition $T = BRB^\top$
14: Return $V = QB$ and $R$.

---

**Algorithm 2** : LanczosNet

1: **Input:** Signal $X$, Lanczos output $V$ and $R$, scale index sets $\mathcal{S}$ and $\mathcal{I}$,
2: **Initialization:** $Y_0 = X$
3: **For** $\ell = 1, 2, \dots, \ell_c$:
4:     $Z = Y_{\ell-1}$, $\mathcal{Z} = \{\emptyset\}$
5:     **For** $j = 1, 2, \dots, \max(\mathcal{S})$:
6:         $Z = SZ$
7:         **If** $j \in \mathcal{S}$:
8:             $\mathcal{Z} = \mathcal{Z} \cup Z$
9:     **For** $i \in \mathcal{I}$:
10:         $\mathcal{Z} = \mathcal{Z} \cup V\hat{R}(\mathcal{I}_i)V^\top Y_{\ell-1}$
11:     $Y_\ell = \text{concat}(\mathcal{Z})W_\ell$
12:     **If** $\ell < L$:
13:         $Y_\ell = \text{Dropout}(\sigma(Y_\ell))$
14: Return $Y_{\ell_c}$.

---

### 3.1 LANCZOS ALGORITHM

Given the aforementioned affinity matrix $S$[1] and node features $x \in \mathbb{R}^{N \times 1}$, the $N$-step Lanczos algorithm computes an orthogonal matrix $Q$ and a symmetric tridiagonal matrix $T$, such that $Q^\top SQ = T$. We denote $Q = [q_1, \cdots, q_N]$ where column vector $q_i$ is the $i$-th Lanczos vector. $T$ is illustrated as below,

$$T = \begin{bmatrix} \gamma_1 & \beta_1 & & \\ \beta_1 & \ddots & \ddots & \\ & \ddots & \ddots & \beta_{N-1} \\ & & \beta_{N-1} & \gamma_N \end{bmatrix}. \tag{2}$$

One can verify that $Q$ forms an orthonormal basis of the Krylov subspace $\mathcal{K}_N(S, x)$ and the first $K$ columns of $Q$ forms the orthonormal basis of $\mathcal{K}_K(S, x)$. The Lanczos algorithm is shown in detail in Alg. 1. Intuitively, if we investigate the $j$-th column of the system $SQ = QT$ and rearrange terms,

---

[1] When faced with a non-symmetric matrix, one can resort to the Arnoldi algorithm.

we obtain $\beta_j q_{j+1} = Sq_j - \beta_{j-1}q_{j-1} - \gamma_j q_j$, which clearly explains lines 4 to 6 of the pseudocode, i.e., it tries to solve the system in an iterative manner. Note that the most expensive operation in the algorithm is the matrix-vector multiplication in line 4. After obtaining the tridiagonal matrix $T$, we can compute the Ritz values and Ritz vectors which approximate the eigenvalues and eigenvectors of $S$ by diagonalizing the matrix $T$. We only add this step in LanczosNet as we found back-propagating through the eigendecomposition in AdaLanczosNet is not numerically stable.

## 3.2 LANCZOSNET

In this section, we first show the construction of the localized polynomial filter based on the Lanczos algorithm's output and discuss its limitations. Then we explain how to construct the spectral filter using a particular low rank approximation and how to further make the filter learnable. At last, we elaborate how to construct multi-scale graph convolution and build a deep network.

**Localized Polynomial Filter**    For the ease of demonstrating the concept of *Krylov subspace*, we consider a pair of input and output channels $(i, j)$. We denote the input as $X_{:,i} \in \mathbb{R}^{N \times 1}$ and the output as $Y_{:,j} \in \mathbb{R}^{N \times 1}$. Executing the Lanczos algorithm for $K$ steps with the normalized $X_{:,i}$ as the starting vector, one can obtain the orthonormal basis $\tilde{Q}$ of $\mathcal{K}_K(S, X_{:,i})$ and the corresponding tridiagonal matrix $\tilde{T}$. Recall that in the localized polynomial filtering, given the orthonormal basis of $\mathcal{K}_K(S, X_{:,i})$, one can write the graph convolution as

$$Y_j = \tilde{Q}\boldsymbol{w}_{i,j}, \tag{3}$$

where $\tilde{Q} \in \mathbb{R}^{N \times K}$ depends on the $X_{:,i}$ and $\boldsymbol{w}_{i,j} \in \mathbb{R}^{K \times 1}$ is the learnable parameter. This filter has the benefit that the corresponding learnable coefficients are compact due to the orthonormal basis. However, if one wants to stack multiple graph convolution layers, the dependency of $\tilde{Q}$ on $X_{:,i}$ implies that a separate run of Lanczos algorithm is necessary for each graph convolution layer which is computationally demanding.

**Spectral Filter**    Ideally, we would like to compute Lanczos vectors only once during the inference of a deep graph convolutional network. Luckily, this can be achieved if we take an alternative view of Lanczos algorithm. In particular, we can choose a random starting vector with unit norm and treat the $K$ step Lanczos layer's output as the low rank approximation $S \approx QTQ^\top$. Note that here $Q \in \mathbb{R}^{N \times K}$ has orthonormal columns and does not depend on the node features $X_i$ and $T$ is a $K \times K$ tridiagonal matrix. Following [14], we prove the theorem below to bound the approximation error.

**Theorem 1.** *Let $U\Lambda U^\top$ be the eigendecomposition of an $N \times N$ symmetric matrix $S$ with $\Lambda_{i,i} = \lambda_i$, $\lambda_1 \geq \cdots \geq \lambda_N$ and $U = [u_1, \ldots, u_N]$. Let $\mathcal{U}_j \equiv \mathrm{span}\{u_1, \ldots, u_j\}$. Assume $K$-step Lanczos algorithm starts with vector $v$ and outputs the orthogonal $Q \in \mathbb{R}^{N \times K}$ and tridiagonal $T \in \mathbb{R}^{K \times K}$. For any $j$ with $1 < j < N$ and $K > j$, we have,*

$$\|S - QTQ^\top\|_F^2 \leq \sum_{i=1}^{j} \lambda_i^2 \left( \frac{\sin(v, \mathcal{U}_i) \prod_{k=1}^{j-1}(\lambda_k - \lambda_N)/(\lambda_k - \lambda_j)}{\cos(v, u_i) T_{K-i}(1 + 2\gamma_i)} \right)^2 + \sum_{i=j+1}^{N} \lambda_i^2,$$

*where $T_{K-i}(x)$ is the Chebyshev Polynomial of degree $K - i$ and $\gamma_i = (\lambda_i - \lambda_{i+1})/(\lambda_{i+1} - \lambda_N)$.*

We leave the proof to the appendix. Note that the term $(\sum_{i=j+1}^{N} \lambda_i^2)^{1/2}$ is the Frobenius norm of the error between $S$ and the best rank-$j$ approximation $S_j$. We decompose the tridiagonal matrix $T = BRB^\top$, where the $K \times K$ diagonal matrix $R$ contains the Ritz values and $B \in \mathbb{R}^{K \times K}$ is an orthogonal matrix. We have a low rank approximation of the affinity matrix $S \approx VRV^\top$, where $V = QB$. Therefore, we can rewrite the graph convolution as,

$$Y_j = [X_i, SX_i, \ldots, S^{K-1}X_i]\boldsymbol{w}_{i,j} \approx [X_i, VRV^\top X_i, \ldots, VR^{K-1}V^\top X_i]\boldsymbol{w}_{i,j}, \tag{4}$$

The difference between Eq. (3) and Eq. (4) is that the former uses the orthonormal basis while the latter uses the approximation of the direct basis of $\mathcal{K}_K(S, X_{:,i})$. Since we explicitly operate on the approximation of spectrum, i.e., Ritz value, it is a spectral filter. Such a filtering form will have significant computational benefits while considering the long range/scale dependency due to the fact that the $t$-th power of $S$ can be approximated as $S^t \approx VR^tV^\top$, where we only need to raise the diagonal entries of $R$ to the power $t$.

**Learning the Spectral Filter**    Following the previous filter, one can naturally design learnable spectral filters. Specifically, we use $K$ different spectral filters of which the $k$-th output $\hat{R}(k) = f_k([R, R^1, \ldots, R^{K-1}])$, where $f_k$ is a multi-layer perceptron (MLP) and $R$ is the diagonal vector of the corresponding diagonal matrix. We then construct a diagonal matrix $\hat{R}(k)$ based on the vector output of $f_k$. Therefore, we have the following filtering,

$$Y_j = [X_i, V\hat{R}(1)V^\top X_i, \ldots, V\hat{R}(K-1)V^\top X_i]\boldsymbol{w}_{i,j}. \tag{5}$$

Note that it includes the polynomial filter as a special case. When positive semi-definiteness is a concern, one can apply an activation function like ReLU to the output of the MLPs.

**Multi-scale Graph Convolution**    Using any above filter, one can construct a deep graph convolutional network which leverages multi-scale information. Taking the learnable spectral filter as an example, we can write one graph convolution layer in a compact way as below,

$$Y = \left[ L^{\mathcal{S}_1}X, \ldots, L^{\mathcal{S}_M}X, V\hat{R}(\mathcal{I}_1)V^\top X, \ldots, V\hat{R}(\mathcal{I}_N)V^\top X \right] W, \tag{6}$$

where weight $W \in \mathbb{R}^{(M+E)D \times O}$, $\mathcal{S}$ is a set of $M$ short scale parameters and $\mathcal{I}$ is a set of $E$ long scale parameters. We consider a non-negative integer as scale parameter, e.g., $\mathcal{S} = \{0, 1, \ldots, 5\}$, $\mathcal{I} = \{10, 20, \ldots, 50\}$. Note that the convolution corresponding to short scales is similar to [8] where the number of matrix-vector multiplications is tied to the maximum scale of $\mathcal{S}$. In contrast, the convolution of long scales decouples the Lanczos step $K$ and scale parameters $\mathcal{I}$, thus permitting great freedom in tuning scales as hyperparameters. One can choose $K$ properly to balance the computation cost and the accuracy of the low rank approximation. In our experiments, short scales are typically less than 10 which have reasonable computation cost. Moreover, the short scale part could sometimes remedy cases where the low rank approximation is crude. We set the long scale no larger than 100 in our experiments. If the maximum eigenvalue of $S$ is 1, we can even raise the power to infinity, which corresponds to the equilibrium state of diffusion process on the graph.

To build a deep network, we can stack multiple such graph convolution layers where each layer has its own spectral filter weights. Nonlinear activation functions, e.g., ReLU, and/or Dropout can be added between layers. The inference algorithm of such a deep network is shown in Alg. 2. With the top layer representation, one can use softmax to perform classification or a fully connected layer to perform regression. The Lanczos algorithm is run beforehand once per graph to construct the network and will not be invoked during inference and learning.

### 3.3    ADALANCZOSNET

In this section, we explain another variant which back-propagates through the Lanczos algorithm. This facilitates learning the graph kernel and/or node embeddings.

**Graph Kernel**    Assume we are given node features $X$ and a graph $\mathcal{G}$. We are interested in learning a graph kernel function with the hope that it can capture the intrinsic geometry of node representations. Given data points $x_i, x_j \in \mathcal{X}$, we define the anisotropic graph kernel, $k : \mathcal{X} \times \mathcal{X} \mapsto \mathbb{R}$ as,

$$k(x_i, x_j) = \exp\left(-\frac{\|(f_\theta(x_i) - f_\theta(x_j))\|^2}{\epsilon}\right). \tag{7}$$

where $f_\theta$ is a MLP. This class of anisotropic kernels is very expressive and includes self-tuning kernel [15] and the Gaussian kernel with Mahalanobis distances [16]. Moreover, for different kernel functions, the resulted graph Laplacians will converge to different limiting operators asymptotically. For example, even for isotropic Gaussian kernels, the graph Laplacian can converge pointwise to the Laplace-Beltrami, Fokker-Planck operator and heat kernel under different normalizations [17, 18]. In practice, we notice that choosing $\epsilon = \sum_{(p,q)\in\mathcal{E}} \|(f_\theta(x_p) - f_\theta(x_q))\|^2/|\mathcal{E}|$ helps normalizing the pairwise distances, thus avoiding the gradient vanishing issue due to the exponential function. This type of learnable anisotropic diffusion is useful in two ways. First, it increases model capacity, thus potentially gaining better performance. Second, it can better adapt to the non-uniform density of the data points on the manifold or nonlinear measurements of the underlying data points on a maninfold. We can construct an adjacency matrix $A$ such that $A_{i,j} = k(x_i, x_j)$ if $(i, j) \in \mathcal{E}$ and $A_{i,j} = 0$ otherwise. Then we can obtain the affinity matrix $S = D^{-\frac{1}{2}}AD^{-\frac{1}{2}}$.

**Node Embedding**    In some applications, we do not observe the node features $X$ but only the graph itself $\mathcal{G}$, so we may need to learn an embedding vector per node. For example, this scenario applies in the quantum chemistry tasks where a node, i.e., an atom within a molecule, has rarely observed features. We can still use the above graph kernel to construct the affinity matrix which results in the same form except $f$ is discarded. Learning embedding $X$ naturally amounts to learning the similarities between nodes.

**Tridiagonal Decomposition**    Although all operations in LanczosNet are differentiable, we empirically observe that backpropagation through the eigendecomposition of the tridiagonal matrix is numerically instable. The situation would be even worse if multiple eigenvalues are numerically close or one takes a large power in Eq. (6). Therefore, we instead directly leverage the approximated tridiagonal decomposition $S \approx QTQ^\top$ which is obtained by running the Lanczos algorithm $K$ steps. Then we can rewrite the graph convolution layer with learnable spectral filter as following,

$$Y = \left[ S^{\mathcal{S}_1} X, \ldots, S^{\mathcal{S}_M} X, Qf_1\left(T^{\mathcal{I}_1}\right) Q^\top X, \ldots, Qf_N\left(T^{\mathcal{I}_N}\right) Q^\top X \right] W, \tag{8}$$

where $f_i$ is a learnable spectral filter. Each $f$ is constructed from a separate MLP denoting as $g$ which takes $T \in \mathbb{R}^{K \times K}$ as input and outputs a same sized matrix. To ensure $f$ outputs a symmetric matrix, we define $f(T) = g(T) + g(T)^\top$.

With the above parameterization of the graph Laplacian and tridiagonal decomposition, we can back-propagate the loss through the Lanczos algorithm to either the graph kernel parameters $\theta$ or the node embedding $X$. The overall model is similar to the LanczosNet except that the Lanczos algorithm needs to be invoked for each inference pass.

## 4    LANCZOS NETWORK AND DIFFUSION MAPS

In this section, we highlight the relationship between LanczosNet and an important example of graph based manifold learning algorithms, diffusion maps [17].

**Diffusion Maps**    In diffusion maps, the weights in the adjacency matrix define a discrete random walk over the graph, where the Markov transition matrix $P = D^{-1}A$ shows the transition probability in a single time step. Therefore, $P_{i,j}^t$ sums the probability of all paths of length $t$ that start at node $i$ and end at node $j$. It is shown in [17] that $P$ can be used to define an inner product in a Hilbert space. Specifically, we use the eigenvalues and right eigenvectors $\{\lambda_l, \psi_l\}_{l=1}^N$ of $P$ to define a diffusion mapping $\Phi_t$ as,

$$\Phi_t(i) = \left( \lambda_1^t \psi_1(i), \lambda_2^t \psi_2(i), \ldots, \lambda_N^t \psi_N(i) \right), \tag{9}$$

where $\psi_l(i)$ is the $i$-th entry of the eigenvector $\psi_l$. Since the row stochastic matrix $P$ is similar to $S$, i.e., $P = D^{-1/2}SD^{1/2}$, we have $\psi_l = D^{-1/2}u_l$. The mapping $\Phi_t$ satisfies $\sum_{k=1}^N P_{i,k}^t P_{j,k}^t / D_{k,k} = \langle \Phi_t(i), \Phi_t(j) \rangle$, where $\langle \cdot, \cdot \rangle$ is the inner product over Euclidean space. The diffusion distance between $i$ and $j$, $d_{\text{DM},t}^2(i,j) = \|\Phi_t(i) - \Phi_t(j)\|^2 = \sum_{k=1}^N (P_{i,k}^t - P_{j,k}^t)^2 / D_{k,k}$, is the weighted-$l_2$ proximity between the probability clouds of random walkers starting at $i$ and ending at $j$ after $t$ steps. Since all eigenvalues of $S$ reside in the interval $[-1,1]$, for some large $t$, $\lambda_l^t$ in Eq. (9) is close to zero, and $d_{\text{DM},t}$ can be well approximated by using only a few largest eigenvalues and their eigenvectors.

**Connection to Graph Convolution**    Apart from using diffusion maps to embed node features $X$ at different time scales, one can use it to compute the frequency representations of $X$ as below,

$$\hat{X} = \Lambda^t U^\top X, \tag{10}$$

where $U$ are the eigenvectors of $S$ and define the graph Fourier transform. The frequency representation $\hat{X}$ is weighted by the powers of the eigenvalues $\lambda_l^t$, suppressing entries with small magnitude of eigenvalues. Recall that in the convolution layer Eq. (5) of LanczosNet, we use multiple such frequency representations with different scales $t$ and replace the eigenvalues $\Lambda$ in Eq. (10) with their approximation, i.e., Ritz values. Therefore, in LanczosNet, spectral filters are actually applied to the frequency representations which are obtained by projecting the node features $X$ onto multiple diffusion maps with different scales.

## 5 RELATED WORK

We can roughly categorize the application of machine learning, especially deep learning, to graph structured data into supervised/semi-supervised and unsupervised scenarios. For the former, a majority of work focuses on node/graph classification and regression [19, 20, 21, 1]. For the latter, unsupervised node/graph embedding learning [22, 23] is common. Recently, generative models for graphs, such as molecule generation, has drawn some attention [24, 25].

**Graph Convolution Based Models** The first class of learning models on graphs stems from graph signal processing (GSP) [12, 4] which tries to generalize convolution operators from traditional signal processing to graphs. Relying on spectral graph theory [26] and graph wavelet theory [27], several definitions of frequency representations of graph signals have been proposed [4]. Among these, spectral graph theory based one is popular, where graph Fourier transform and its inverse are defined based on the eigenbasis of the graph Laplacian. Following this line, many graph convolution based deep network models emerge. [2, 28] are among the first to explore Laplacian based graph convolution within the context of deep networks. Meanwhile, [29] performs graph convolution directly based on the adjacency matrix to predict molecule fingerprints. [30] proposes a strategy to form same sized local neighborhoods and then apply convolution like regular CNNs. Chebyshev polynomials are exploited by [7] to construct localized polynomial filters for graph convolution and are later simplified in graph convolutional networks (GCN) [11]. Further accelerations for GCN based on importance sampling and control variate techniques have been proposed by [31, 32]. Several attention mechanisms have been introduced in [33, 34] to learn the weights over edges for GCNs. Notably, [8] proposes diffusion convolutional neural networks (DCNN) which uses diffusion operator for graph convolution. Lanczos method has been explored for graph convolution in [35] for the purpose of acceleration. Specifically, they only consider the localized polynomial filter case in our LanczosNet variant and do not explore the low rank decomposition, learnable spectral filter and graph kernel/node embedding learning as we do.

**Recurrent Neural Networks based Models** The second class of models dates back to recursive neural networks [36] which recurrently apply neural networks to trees following a particular order. Graph neural networks (GNN) [3] generalize recursive neural networks to arbitrary graphs and exploit the synchronous schedule to propagate information on graphs. [37] later proposes the gated graph neural networks (GGNN) which improves GNN by adding gated recurrent unit and training the network with back-propagation through time. [38] learns graph embeddings via unrolling variational inference algorithms over a graph as a RNN. [39] introduces random subgraph sampling and explores different aggregation functions to scale GNN to large graphs. [40] proposes asynchronous propagation schedules based on graph partitions to improve GNN. Moreover, many applications have recently emerged for GNNs, including community detection [41], situation recognition [42], RGBD semantic segmentation [43], few-shot learning [21], probabilistic inference [44], continuous control of reinforcement learning [45, 46] and so on.

**Graph based Manifold Learning** The non-linear dimensionality reduction methods, such as locally linear embedding (LLE) [47], ISOMAP [48], Hessian LLE [49], Laplacian eigenmaps [50], and diffusion maps [17], assume that the high-dimensional data lie on or close to a low dimensional manifold and use the local affinities in the weighted graph to learn the global features of the data. They are invaluable tools for embedding complex data in a low dimensional space and regressing functions over graphs. Spectral clustering [51, 52], semi-supervised learning [53], and out-of-sample extension [54] share the similar geometrical consideration of the associated graphs. Anisotropic graph kernels are useful in many applications. For example, [15] improves the spectral clustering results with a self-tuning diffusion kernel that takes into account the local variance at each node in the Gaussian kernel function. Similarly, [55] uses the anisotropic Gaussian kernel defined by the local Mahalanobis distances to extract independent components from nonlinear measurements of independent stochastic Itô processes. Manifold learning with anisotropic kernel is also useful for data-driven dynamical system analysis, for example, detecting intrinsically slow variable for a stochastic dynamical system [56], filtering dynamical processes [57], and long range climate forecasting [58, 59]. The anisotropic diffusion is able to use the local statistics of the measurements to convey the geometric information on the underlying factors rather than the specific realization or measurements at hand [60, 61].

| Cora | GCN-FP | GGNN | DCNN | ChebyNet | GCN | MPNN | GraphSAGE | GAT | LNet | AdaLNet |
|---|---|---|---|---|---|---|---|---|---|---|
| Public | 74.6 ± 0.7 | 77.6 ± 1.7 | 79.7 ± 0.8 | 78.0 ± 1.2 | 80.5 ± 0.8 | 78.0 ± 1.1 | 74.5 ± 0.8 | **82.6 ± 0.7** | 79.5 ± 1.8 | 80.4 ± 1.1 |
| 3% | 71.7 ± 2.4 | 73.1 ± 2.3 | 76.7 ± 2.5 | 62.1 ± 6.7 | 74.0 ± 2.8 | 72.0 ± 4.6 | 64.2 ± 4.0 | 56.8 ± 7.9 | 76.3 ± 2.3 | **77.7 ± 2.4** |
| 1% | 59.6 ± 6.5 | 60.5 ± 7.1 | 66.4 ± 8.2 | 44.2 ± 5.6 | 61.0 ± 7.2 | 56.7 ± 5.9 | 49.0 ± 5.8 | 48.6 ± 8.0 | 66.1 ± 8.2 | **67.5 ± 8.7** |
| 0.5% | 50.5 ± 6.0 | 48.2 ± 5.7 | 59.0 ± 10.7 | 33.9 ± 5.0 | 52.9 ± 7.4 | 46.5 ± 7.5 | 37.5 ± 5.4 | 41.4 ± 6.9 | 58.1 ± 8.2 | **60.8 ± 9.0** |

| Citeseer | GCN-FP | GGNN | DCNN | ChebyNet | GCN | MPNN | GraphSAGE | GAT | LNet | AdaLNet |
|---|---|---|---|---|---|---|---|---|---|---|
| Public | 61.5 ± 0.9 | 64.6 ± 1.3 | 69.4 ± 1.3 | 70.1 ± 0.8 | 68.1 ± 1.3 | 64.0 ± 1.9 | 67.2 ± 1.0 | **72.2 ± 0.9** | 66.2 ± 1.9 | 68.7 ± 1.0 |
| 1% | 54.3 ± 4.4 | 56.0 ± 3.4 | 62.2 ± 2.5 | 59.4 ± 5.4 | 58.3 ± 4.0 | 54.3 ± 3.5 | 51.0 ± 5.7 | 46.5 ± 9.3 | 61.3 ± 3.9 | **63.3 ± 1.8** |
| 0.5% | 43.9 ± 4.2 | 44.3 ± 3.8 | 53.1 ± 4.4 | 45.3 ± 6.6 | 47.7 ± 4.4 | 41.8 ± 5.0 | 33.8 ± 7.0 | 38.2 ± 7.1 | 53.2 ± 4.0 | **53.8 ± 4.7** |
| 0.3% | 38.4 ± 5.8 | 36.5 ± 5.1 | 44.3 ± 5.1 | 39.3 ± 4.9 | 39.2 ± 6.3 | 36.0 ± 6.1 | 25.7 ± 6.1 | 30.9 ± 6.9 | 44.4 ± 4.5 | **46.7 ± 5.6** |

| Pubmed | GCN-FP | GGNN | DCNN | ChebyNet | GCN | MPNN | GraphSAGE | GAT | LNet | AdaLNet |
|---|---|---|---|---|---|---|---|---|---|---|
| Public | 76.0 ± 0.7 | 75.8 ± 0.9 | 76.8 ± 0.8 | 69.8 ± 1.1 | 77.8 ± 0.7 | 75.6 ± 1.0 | 76.8 ± 0.6 | 76.7 +- 0.5 | **78.3 ± 0.3** | 78.1 ± 0.4 |
| 0.1% | 70.3 ± 4.7 | 70.4 ± 4.5 | 73.1 ± 4.7 | 55.2 ± 6.8 | 73.0 ± 5.5 | 67.3 ± 4.7 | 65.4 ± 6.2 | 59.6 +- 9.5 | **73.4 ± 5.1** | 72.8 ± 4.6 |
| 0.05% | 63.2 ± 4.7 | 63.3 ± 4.0 | 66.7 ± 5.3 | 48.2 ± 7.4 | 64.6 ± 7.5 | 59.6 ± 4.0 | 53.0 ± 8.0 | 50.4 +- 9.7 | **68.8 ± 5.6** | 66.0 ± 4.5 |
| 0.03% | 56.2 ± 7.7 | 55.8 ± 7.7 | 60.9 ± 8.2 | 45.3 ± 4.5 | 57.9 ± 8.1 | 53.9 ± 6.9 | 45.4 ± 5.5 | 50.9 +- 8.8 | 60.4 ± 8.6 | **61.0 ± 8.7** |

Table 1: Test accuracy with 10 runs on citation networks. The public splits in Cora, Citeseer and Pubmed contain 5.2%, 3.6% and 0.3% training examples respectively.

## 6 EXPERIMENTS

In this section, we compare our two model variants against 9 recent graph networks, including graph convolution networks for fingerprint (GCN-FP) [29], gated graph neural networks (GGNN) [37], diffusion convolutional neural networks (DCNN) [8], Chebyshev networks (ChebyNet) [7], graph convolutional networks (GCN) [11], message passing neural networks (MPNN) [62], graph sample and aggregate (GraphSAGE) [39], graph partition neural networks (GPNN) [40], graph attention networks (GAT) [33]. We test them on two sets of tasks: (1) semi-supervised document classification on 3 citation networks [63], (2) supervised regression of molecule property on QM8 quantum chemistry dataset [64]. For fair comparison, we only tune model-related hyperparameters in all our experiments and share the others, e.g., using the same batch size. We carefully tune hyperparameters based on cross-validation and report the best performance of each competitor. Please refer to the appendix for more details on hyperparameters. We implement all methods using PyTorch [65] and release the code at https://github.com/lrjconan/LanczosNetwork.

### 6.1 CITATION NETWORKS

Three citation networks used in this experiment are: Cora, Citeseer and Pubmed. For each network, nodes are documents and connected based on their citation links. Each node is associated with a bag-of-words feature vector. We use the same pre-processing procedure and follow the transductive setting as in [63]. In particular, given a portion of nodes and their labeled content categories, e.g., history, science, the task is to predict the category for other unlabeled nodes within the same graph. The statistics of these datasets are summarized in the appendix. All experiments are repeated 10 times with different random seeds. During each run, all methods share the same random seed. We first experiment with the public data split and observe severe overfitting for almost all algorithms. To mitigate overfitting and test the robustness of models, we then increase the difficulty of the task by reducing the portion of training examples to several levels and randomly split data.

Experimental results and exact portions of training examples are shown in Table. 1. We use the reported best hyperparameters when available for public split and do cross-validation otherwise. Hyperparameters are reported in the appendix. From the table, we see that for random splits with different portion of training examples, since each run of experiment uses a separate random split, the overall variance is larger than its public counterpart. We see that GAT achieves the best performance on the public split but performs poorly on random splits with different portions of training examples. This is partly due to the fact that GAT uses multiple dropout throughout the model which helps only if there is overfitting. We can see that either LanczosNet or AdaLanczosNet achieves state-of-the-art accuracy on random difficult splits and performs closely with respect to GAT on public splits. This may be attributed to the fact that with fewer training examples, the model requires longer scale schemes to spread supervised information over the graph. Our model provides an efficient way of leveraging such long scale information.

| Methods | Validation MAE ($\times 1.0e^{-3}$) | Test MAE ($\times 1.0e^{-3}$) |
|---|---|---|
| GCN-FP [29] | $15.06 \pm 0.04$ | $14.80 \pm 0.09$ |
| GGNN [37] | $12.94 \pm 0.05$ | $12.67 \pm 0.22$ |
| DCNN [8] | $10.14 \pm 0.05$ | $9.97 \pm 0.09$ |
| ChebyNet [7] | $10.24 \pm 0.06$ | $10.07 \pm 0.09$ |
| GCN [11] | $11.68 \pm 0.09$ | $11.41 \pm 0.10$ |
| MPNN [62] | $11.16 \pm 0.13$ | $11.08 \pm 0.11$ |
| GraphSAGE [39] | $13.19 \pm 0.04$ | $12.95 \pm 0.11$ |
| GPNN [40] | $12.81 \pm 0.80$ | $12.39 \pm 0.77$ |
| GAT [33] | $11.39 \pm 0.09$ | $11.02 \pm 0.06$ |
| LanczosNet | $\mathbf{9.65 \pm 0.19}$ | $\mathbf{9.58 \pm 0.14}$ |
| AdaLanczosNet | $10.10 \pm 0.22$ | $9.97 \pm 0.20$ |

Table 2: Mean absolute error on QM8 dataset.

## 6.2 QUANTUM CHEMISTRY

We then benchmark all algorithms on the QM8 quantum chemistry dataset which comes from a recent study on modeling quantum mechanical calculations of electronic spectra and excited state energy of small molecules [64]. The setup of QM8 is as follows. Atoms are treated as nodes and they are connected to each other following the structure of the corresponding molecule. Each edge is labeled with a chemical bond. Note that two atoms in one molecule can have multiple edges belong to different chemical bonds. Therefore a molecule is actually modeled as a multigraph. We also use explicit hydrogen in molecule graphs as suggested in [62]. Since some models cannot leverage feature on edges easily, we use the molecule graph itself as the only input information for all models so that it is a fair comparison. As demonstrated in our ablation studies, learning node embeddings for atoms is very helpful. Therefore, we augment all competitors and our models with this component. The task is to predict 16 different quantities of electronic spectra and energy per molecule graph which boils down to a regression problem. There are 21786 molecule graphs in total of which the average numbers of nodes and edges are around 16 and 21. There are 6 different chemical bonds and 70 different atoms throughout the dataset. We use the split provided by DeepChem [2] which have 17428, 2179 and 2179 graphs for training, validation and testing respectively. Following [62, 66], we use mean squared error (MSE) as the loss for training and weighted mean absolute error (MAE) as the evaluation metric. We repeat all experiments 3 times with different random seeds and report the average performance and standard deviation. The same random seed is shared for all methods per run. Hyperparameters are reported in the appendix. The validation and test MAE of all methods are shown in Table 2. As you can see, LanczosNet and AdaLanczosNet achieve better performances than all other competitors. Note that DCNN also achieves good performance with the carefully chosen scale parameters since it is somewhat similar to our model in terms of leveraging multi-scale information.

## 6.3 ABLATION STUDY

We also did a thorough ablation study of our modeling components on the validation set of QM8.

**Multi-Scale Graph Convolution:** We first study the effect of multi-scale graph convolution. In order to rule out the impact of other factors, we use LanczosNet, do not employ the learnable spectral filter and use the one-hot encoding as the node embedding. The results are shown in the first row of Table 3. Using long scales for graph convolution clearly helps on this task. Combining both short and long scales further improves results.

**Lanczos Step:** We then investigate the Lanczos step since it will have an impact on the accuracy of the low rank approximation induced by the Lanczos algorithm. The results are shown in the second row of Table 3. We can see that the performance is better with a relatively small Lanczos step like 10 and 20 which makes sense since the average number of nodes in this dataset is around 16.

**Learning Spectral Filter:** We then study whether learning spectral filter will help improve performance. The results are shown in the third row of Table 3. Adding a 3-layer MLP does help reduce the

---

[2]https://deepchem.io/

| Model | Graph Kernel | Node Embedding | Spectral Filter | Short Scales | Long Scales | Lanczos Step | Validation MAE ($\times 1.0e^{-3}$) |
|---|---|---|---|---|---|---|---|
| LanczosNet | | one-hot | | {1, 2, 3} | | | 10.71 |
| LanczosNet | | one-hot | | {3, 5, 7} | | | 10.60 |
| LanczosNet | | one-hot | | | {10, 20, 30} | 20 | 10.54 |
| LanczosNet | | one-hot | | {3, 5 ,7} | {10, 20, 30} | 20 | **10.41** |
| LanczosNet | | one-hot | | | {10, 20, 30} | 5 | 10.49 |
| LanczosNet | | one-hot | | | {10, 20, 30} | 10 | **10.44** |
| LanczosNet | | one-hot | | | {10, 20, 30} | 20 | 10.54 |
| LanczosNet | | one-hot | | | {10, 20, 30} | 40 | 10.49 |
| LanczosNet | | one-hot | 3-MLP | {3, 5 ,7} | {10, 20, 30} | 20 | 10.44 |
| LanczosNet | | one-hot | 5-MLP | {3, 5 ,7} | {10, 20, 30} | 20 | 10.54 |
| LanczosNet | | ✓ | 3-MLP | {3, 5 ,7} | {10, 20, 30} | 20 | 10.26 |
| LanczosNet | | ✓ | 3-MLP | | {1, 2, 3, 5, 7, 10, 20, 30} | 20 | **9.56** |
| AdaLanczosNet | ✓ | one-hot | 3-MLP | {3, 5, 7} | {10, 20, 30} | 20 | 10.99 |
| AdaLanczosNet | | ✓ | 3-MLP | {3, 5, 7} | {10, 20, 30} | 20 | 10.20 |
| AdaLanczosNet | | ✓ | 3-MLP | {1, 2, 3} | {5, 7, 10, 20, 30} | 20 | **9.96** |

Table 3: Ablation study on QM8 dataset. Empty cell means that the component is neither used nor applicable. $X$-MLP means a MLP with $X$ hidden layers. 'one-hot' means the node embedding is fixed as the one-hot encoding throughout learning and inference.

error compared to not using any learnable spectral filter. Note that the MLP consists of 128 hidden units per layer and uses ReLU as the nonlinearity. However, using a deeper MLP does not seem to be helpful which might be caused by the challenges in optimization.

**Graph Kernel/Node Embedding:** At last, we study the usefulness of adding graph kernel and node embeddings. We first fix the node embedding as one-hot encoding and learn a 3 layer MLP which is the function $f_\theta$ in Eq. (7). Next, we learn the node embeddings directly. Intuitively, learning embeddings amounts to learn a separate function $f$ per node whereas our graph kernel learning enforces that $f$ is shared for all nodes, thus being more restrictive. As shown in the 3-rd and 4-th rows of the table, learning node embeddings significantly improves the performance for both LanczosNet and AdaLanczosNet and is more effective than learning graph kernels. Also, tuning the scale parameters further boosts the performance.

## 7 CONCLUSION

In this paper, we propose LanczosNet which leverages the Lanczos algorithm to construct a low rank approximation of the graph Laplacian. It not only provides an efficient way to gather multi-scale information for graph convolution but also enables learning spectral filters. Additionally, we propose a model variant AdaLanczosNet which facilitates graph kernel and node embedding learning. We show that our model has a close relationship with graph based manifold learning, especially diffusion map. Experimental results demonstrate that our model outperforms a range of other graph networks, on challenging graph problems. We are currently exploring customized eigen-decomposition methods for tridiagonal matrices, which will potentially further improve our AdaLanczosNet. Overall, work in this direction holds promise for allowing deep learning to scale up to very large graph problems.

## ACKNOWLEDGMENTS

RL thanks Roger Grosse for introducing the Lanczos algorithm to him. RL was supported by Connaught International Scholarships. RL, RU and RZ were supported in part by the Intelligence Advanced Research Projects Activity (IARPA) via Department of Interior/Interior Business Center (DoI/IBC) contract number D16PC00003. The U.S. Government is authorized to reproduce and distribute reprints for Governmental purposes notwithstanding any copyright annotation thereon. Disclaimer: the views and conclusions contained herein are those of the authors and should not be interpreted as necessarily representing the official policies or endorsements, either expressed or implied, of IARPA, DoI/IBC, or the U.S. Government.

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

## 8 APPENDIX

### 8.1 LOW RANK APPROXIMATION

We first state the following Lemma from [67] without proof and then prove our Theorem 1 following [14].

**Lemma 1.** *Let $A \in \mathbb{R}^{N \times N}$ be symmetric and $v$ an arbitrary vector. Define Krylov subspace $\mathcal{K}_m \equiv span\{v, Av, \ldots, A^{m-1}v\}$. Let $A = U\Lambda U^{\top}$ be the eigendecomposition of $A$ with $\Lambda_{i,i} = \lambda_i$ and $\lambda_1 \geq \cdots \geq \lambda_n$. Denoting $U = [u_1, \ldots, u_N]$ and $\mathcal{U}_j = \text{span}\{u_1, \ldots, u_j\}$, then*

$$\tan(u_j, \mathcal{K}_m) \leq \frac{\sin(v, \mathcal{U}_j) \prod_{k=1}^{j-1} (\lambda_k - \lambda_n)/(\lambda_k - \lambda_j)}{\cos(v, u_j) T_{m-j}(1 + 2\gamma)},$$

*where $T_{m-j}(x)$ is the Chebyshev Polynomial of degree $m - j$ and $\gamma = (\lambda_j - \lambda_{j+1})/(\lambda_{j+1} - \lambda_N)$.*

**Theorem 1.** *Let $U\Lambda U^{\top}$ be the eigendecomposition of an $N \times N$ symmetric matrix $S$ with $\Lambda_{i,i} = \lambda_i$, $\lambda_1 \geq \cdots \geq \lambda_N$ and $U = [u_1, \ldots, u_N]$. Let $\mathcal{U}_j \equiv \text{span}\{u_1, \ldots, u_j\}$. Assume $K$-step Lanczos algorithm starts with vector $v$ and outputs the orthogonal $Q \in \mathbb{R}^{N \times K}$ and tridiagonal $T \in \mathbb{R}^{K \times K}$. For any $j$ with $1 < j < N$ and $K > j$, we have,*

$$\|S - QTQ^{\top}\|_F^2 \leq \sum_{i=1}^{j} \lambda_i^2 \left( \frac{\sin(v, \mathcal{U}_i) \prod_{k=1}^{j-1} (\lambda_k - \lambda_N)/(\lambda_k - \lambda_j)}{\cos(v, u_i) T_{K-i}(1 + 2\gamma_i)} \right)^2 + \sum_{i=j+1}^{N} \lambda_i^2,$$

*where $T_{K-i}(x)$ is the Chebyshev Polynomial of degree $K - i$ and $\gamma_i = (\lambda_i - \lambda_{i+1})/(\lambda_{i+1} - \lambda_N)$.*

*Proof.* From Lanczos algorithm, we have $SQ = QT$. Therefore,

$$\|S - QTQ^{\top}\|_F^2 = \|S - SQQ^{\top}\|_F^2$$
$$= \|S(I - QQ^{\top})\|_F^2 \tag{11}$$

Let $P_Q^{\perp} \equiv I - QQ^{\top}$, the orthogonal projection onto the orthogonal complement of subspace span$\{Q\}$. Relying on the eigendecomposition, we have,

$$\|S - QTQ^{\top}\|_F^2 = \|U\Lambda U^{\top}(I - QQ^{\top})\|_F^2$$
$$= \|\Lambda U^{\top}(I - QQ^{\top})\|_F^2$$
$$= \|(I - QQ^{\top})U\Lambda\|_F^2$$
$$= \| [\lambda_1 P_Q^{\perp} u_1, \ldots, \lambda_N P_Q^{\perp} u_N] \|_F^2, \tag{12}$$

where we use the fact that $\|RA\|_F^2 = \|A\|_F^2$ for any orthogonal matrix $R$ and $\|A^{\top}\|_F^2 = \|A\|_F^2$.

Note that for any $j$ we have,

$$\| [\lambda_1 P_Q^{\perp} u_1, \ldots, \lambda_N P_Q^{\perp} u_N] \|_F^2 = \sum_{i=1}^{N} \lambda_i^2 \|P_Q^{\perp} u_i\|^2$$
$$\leq \sum_{i=1}^{j} \lambda_i^2 \|P_Q^{\perp} u_i\|^2 + \sum_{i=j+1}^{N} \lambda_i^2, \tag{13}$$

where we use the fact that for any $i$, $\|P_Q^{\perp} u_i\|^2 = \|u_i\|^2 - \|u_i - P_Q^{\perp} u_i\|^2 \leq \|u_i\|^2 = 1$.

Note that we have span$\{Q\}$ = span$\{v, Sv, \ldots, S^{K-1}v\} \equiv \mathcal{K}_K$ from the Lanczos algorithm. Therefore, we have,

$$\|P_Q^{\perp} u_i\| = |\sin(u_i, \mathcal{K}_K)| \leq |\tan(u_i, \mathcal{K}_K)|. \tag{14}$$

Applying the above lemma with $A = S$, we finish the proof.

$$\square$$

| Dataset | #Nodes | #Edges | #Classes | #Features | $S_0$ | $S_1$ | $S_2$ | $S_3$ |
|---------|--------|--------|----------|-----------|-------|-------|-------|-------|
| Citeseer | 3,327 | 4,732 | 6 | 3,703 | 3.6% | 3% | 1% | 0.5% |
| Cora | 2,708 | 5,429 | 7 | 1,433 | 5.2% | 1% | 0.5% | 0.3% |
| Pubmed | 19,717 | 44,338 | 3 | 500 | 0.3% | 0.1% | 0.05% | 0.03% |

Table 4: Dataset statistics. $S_0$ is portion of training examples in the public split. $S_1$ to $S_3$ are the ones of 3 random splits generated by us.

## 8.2 LANCZOS ALGORITHM

Utilizing exact arithmetic, Lanczos vectors are orthogonal to each other. However, in floating point arithmetic, it is well known that the round-off error will make the Lanczos vectors lose orthogonality as the iteration proceeds. One could apply a full Gram-Schmidt (GS) process $z = z - \sum_{i=1}^{j-1} z^\top q_i q_i$ after line 6 of Alg. 1 to ensure orthogonality. Other partial or selective re-orthogonalization could also be explored. However, since we found the orthgonality issue does not hurt overall performance with a small iteration number, e.g., $K = 20$, and the full GS process is computationally expensive, we do not add such a step. Although some customized eigendecomposition methods, e.g., implicit QL [68], exist for tridiagonal matrix, we leave it for future exploration due to its complicated implementation.

## 8.3 EXPERIMENTS

For ChebyNet, we do not use graph coarsening in all experiments due to its demanding computation cost for large graphs. Also, for small molecule graphs, coarsening generally does not help since it loses information compared to directly stacking another layer of original graph.

**Citation Networks**    The statistics of three citation networks are summarized in Table 4. We now report the important hyperparameters chosen via cross-validation for each method. All methods are trained with Adam with learning rate $1.0e^{-2}$ and weight decay $5.0e^{-4}$. The maximum number of epoch is set to 200. Early stop with window size 10 is also adopted. We tune hyperparameters using Cora alone and fix them for citeseer and pubmed. For convolution based methods, we found 2 layers work the best. In GCN-FP, we set the hidden dimension to 64 and dropout to 0.5. In GGNN, we set the hidden dimension to 64, the propagate step to 2 and aggregation function to summation. In DCNN, we set the hidden dimension to 64, dropout to 0.5 and use diffusion scales $\{1, 2, 5\}$. In ChebyNet, we set the polynomial order to 5, the hidden dimension to 64 and dropout to 0.5. In GCN, we set the hidden dimension to 64 and dropout to 0.5. In MPNN, we use GRU as update function and set the hidden dimension to 64 and dropout to 0.5. No edge embedding is used as there is just one edge type. In GraphSAGE, we set the number of sampled neighbors to 500, the hidden dimension to 64, dropout to 0.5 and the aggregation function to average. In GAT, we set the number of heads per layer to 8, 1, hidden dimension per head to 8 and dropout to 0.6. In LanczosNet, we set the short and long diffusion scales to $\{1, 2, 5, 7\}$ and $\{10, 20, 30\}$ respectively. The hidden dimension is 64 and dropout is 0.5. Lanczos step is 20. 1-layer MLP with 128 hidden units and ReLU nonlinearity is used as the spectral filter. In AdaLanczosNet, we set the short and long diffusion scales to $\{1, 2, 5\}$ and $\{10, 20\}$ respectively. The hidden dimension is 64 and dropout is 0.5. Lanczos step is 20. 1-layer MLP with 128 hidden units and ReLU nonlinearity is used as the spectral filter.

**Quantum Chemistry**    We now report the important hyperparameters chosen via cross-validation for each method. All methods are trained with Adam with learning rate $1.0e^{-4}$ and no weight decay. The maximum number of epoch is set to 200. Early stop with window size 10 is also adopted. For convolution based methods, we found 7 layers work the best. We augment all methods with 64-dimension node embedding and add edge types by either feeding a multiple-channel graph Laplacian matrix or directly adding a separate message function per edge type. For all methods, no dropout is used since it slightly hurts the performance. In GCN-FP, we set the hidden dimension to 128. In GGNN, we set the hidden dimension to 128, the propagate step to 15 and aggregation function to average. In DCNN, we set the hidden dimension to 128 and use diffusion scales $\{3, 5, 7, 10, 20, 30\}$. In ChebyNet, we set the polynomial order to 5, the hidden dimension to 128. In GCN, we set the hidden dimension to 128. In MPNN, we use GRU as update function, set the number of propagation to 7, set the hidden dimension to 128, use a 1-layer MLP with 1024 hidden units and ReLU nonlinearity

as the message function and set the number of unroll step of *Set2Vec* to 10. In GraphSAGE, we set the number of sampled neighbors to 40, the hidden dimension to 128 and the aggregation function to average. In GAT, we set the number of heads of all 7 layers to 8 and hidden dimension per head to 16. In LanczosNet, we do not use short diffusion scales and set long ones to $\{1, 2, 3, 5, 7, 10, 20, 30\}$. The hidden dimension is 128. Lanczos step is 20. 1-layer MLP with 128 hidden units and ReLU nonlinearity is used as the spectral filter. In AdaLanczosNet, we set the short and long diffusion scales to $\{1, 2, 3\}$ and $\{5, 7, 10, 20, 30\}$ respectively. The hidden dimension is 128. Lanczos step is 20. 3-layer MLP with 4096 hidden units and ReLU nonlinearity is used as the spectral filter.

