# OpenReview forum: "LanczosNet: Multi-Scale Deep Graph Convolutional Networks"
_ICLR.cc/2019/Conference_

### Official Review · AnonReviewer2 · 2018-10-31
**Novel approach to graph neural networks with strong empirical evaluation**

**Rating:** 8
**Confidence:** 4

**Review:**

The authors propose a novel method for learning graph convolutional networks. The core idea is to use the Lanczos algorithm to obtain a low-rank approximation of the graph Laplacian. The authors propose two ways to include the Lanczos algorithm. First, as a preprocessing step where the algorithm is applied once on the input graph and the resulting approximation is fixed during learning. Second, by including a differentiable version of the algorithm into an end-to-end trainable model.

The proposed method is novel and achieves good results on a set of experiments.

The authors discuss related work in a thorough and meaningful manner.

There is not much to criticize. This is a very good paper. The almost 10 pages are perhaps a bit excessive considering there was an (informal) 8 page limit. It might make sense to provide a more accessible discussion of the method and Theorem 1, and move some more detailed/technical parts in pages 4, 5, and 6 to an appendix.

---

> ### Author Response · Authors · 2018-11-20
> **Response**
>
> Thanks for the comments! We will improve the writing and make the main contributions more clear.

---

### Official Review · AnonReviewer3 · 2018-11-05
**interesting ideas, but sometimes all over the place**

**Rating:** 7
**Confidence:** 5

**Review:**

This paper proposes to use a Lanczos alogrithm, to get approximate decompositions of the graph Laplacian, which would facilitate the computation and learning of spectral features in graph convnets. It further proposes an extension with back propagation through the Lanczos algorithm, in order to train end to end models.

Overall, the idea of using Lanczos algorithm to bypass the computation of the eigendecomposition, and thus simplify filtering operations in graph signal processing is not new [e.g., 35]. However, using this algorithm in the framework of graph convents is new, and certainly interesting. The authors seem to claim that their method permits to learn spectral filters, what other methods could not do - this is not completely true and should probably be rephrased more clearly: many graph convnets, actually learn features.

The general construction and presentation of the algorithms are generally clear, and pretty complete. A few things that could be clarified are the following:

- in the spectral filters of Eq (4), what gets fundamentally different from polynomial filters proposed in other graph convnets architectures?
- what happens when the graph change? Do the learned features make sense on different graphs? And if yes, why? If not, the authors should be more explicit in their presentation
- what is the complexity of the proposed methods? that should be minimally discussed (at least), as it is part of the key motivations for the proposed algorithms
- how is the learning done in 3.2? If there is any learning at all? (btw, S below Eq (6) is a poor notation choice, as S is used earlier for something else)
- the results are not very impressive - they are good, but not stellar, and could benefit from showing an explicit tradeoff in terms of complexity too?

The discussion in the related work, and the analogy with manifold learning are interesting. However, that brings probably to one of the main issues with the papers - the authors are obviously very knowledgeable in graph convnets, graph signal processing, and optimisation. However, there are really too many things in this paper, which leads to numerous shortcuts, and some time confusion. Given the page limits, not everything can be treated with the level of details that it would deserve. It might be good to consider trimming down the paper to its main and core aspects for the next version.

---

> ### Author Response · Authors · 2018-11-20
> **Response**
>
> Thanks for the careful reading and the constructive comments! We will improve the writing and make the paper more accessible in terms of main contributions. Additionally, we would like to clarify a few raised questions as below.
>
> Q1: What gets fundamentally different from polynomial filters proposed in other graph convnets architectures?
>
> A1: We mainly compare with the Chebyshev polynomial filter since it is the most frequently used and also has the nice orthogonality property.
>
> First, Chebyshev polynomial filters can be regarded as a special case of our learnable spectral filters. The expansion of the Chebyshev recursion manifests that the filtering lies in a Krylov subspace of which the eigenbasis can be achieved by Lanczos algorithm. Therefore, recovering Chebyshev polynomial filters reduces to recovering the specific coefficients of polynomials which can be achieved by a multi-layer perceptron (MLP) due to its universal approximation power.
>
> Second, we decouple the order of polynomial and the number of eigenbasis which is not the case for Chebyshev polynomial. Recall that computing K-th order Chebyshev polynomial, i.e., finding K basis vectors, requires running the recursion K times. However, we can run the Lanczos algorithm for M steps, e.g., M < K, to get M basis vectors. Then we can easily get the K-th order polynomial by directly raising the K-th power of Ritz values.
>
> We will discuss more on this difference in our later version.
>
> Q2: What happens when the graph change? Do the learned features make sense on different graphs? And if yes, why? If not, the authors should be more explicit in their presentation.
>
> A2: Like many other graph convolutional networks, learnable parameters of our model do not depend on any graph specific quantities, like the number of nodes or edges, thus permitting generalization over different graphs. Moreover, in our QM8 experiments, different molecules are indeed different graphs. Therefore, the experimental results empirically verify that our learned features can generalize to different graphs. In terms of why they generalize, we currently do not have a satisfying answer as it requires deep understanding of the data distribution, model expressiveness and non-trivial inequality techniques for proving a useful generalization bound. Intuitively, the successful generalization may be due to the fact that our model does capture some patterns of sub-graphs within the molecules. These patterns may frequently appear in different molecules and determine the physical and chemical properties which link to the final predicted energy. We will improve our presentation regarding to this point.
>
> Q3: What is the complexity of the proposed methods? that should be minimally discussed (at least), as it is part of the key motivations for the proposed algorithms.
>
> A3: It is hard to describe the overall time complexity in a concise manner as it requires lengthy notation. For the Lanczos algorithm alone, assuming the graph has N nodes, the most computationally expensive operation of our Algorithm 1 is the matrix vector product in line 4 which generally costs O(N^2) per step. If we further assume the algorithm runs for K steps, then the overall time complexity is O(K(N^2)). It is economical since a single graph convolution operation in any graph convnets is also generally O(N^2). In contrast, the eigen decomposition is generally O(N^3). We will discuss this in the later version.
>
> Q4: How is the learning done in 3.2? If there is any learning at all? (btw, S below Eq (6) is a poor notation choice, as S is used earlier for something else).
>
> A4: For the spectral filter, the learning is done via learning the MLP which maps Ritz values R to R_hat, i.e., f as described above Eq. (5). S below Eq (6) is actually in different font style. We will change the notation to improve the presentation.
>
> Q5: The results are not very impressive - they are good, but not stellar, and could benefit from showing an explicit tradeoff in terms of complexity too?
>
> A5: We have partially updated experimental results by adding spectral filters in a layer-wise manner. Please refer to our common response. We will also show the run-time in the later version to contrast these methods.

---

### Official Review · AnonReviewer1 · 2018-11-06
**Paper brings insights and develops novel techniques for graph convolutional networks based on the Lanczos algorithm.**

**Rating:** 7
**Confidence:** 3

**Review:**

The paper under review builds useful insights and novel methods for graph convolutional networks, based on the Lanczos algorithm for efficient computations involving the graph Laplacian matrices induced by the neighbor edge structure of graph networks.

While previous work [35] has explored the Lanczos algorithm from numerical linear algebra as a means to accelerate computations in graph convolutional networks, the current paper goes further by:
(1) exploring in significant more depth the low rank decomposition underlying the Lanczos algorithm.
(2) learning the spectral filter (beyond the Chebychev design) and potentially also the graph kernel and node embedding.
(3) drawing interesting connections with graph diffusion methods which naturally arise from the matrix power computation inherent to the Lanczos iteration.

The paper includes a systematic evaluation of the proposed approach and comparison with existing methods on two tasks: semi-supervised learning in citation networks and molecule property prediction from interactions in atom networks. The main advantage of the proposed method as illustrated in particular by the experimental results in the citation network domain is its ability to generalize well in the presence of a small  amount of training data, which the authors attribute to its efficient capturing of both short- and long-range interactions.

In terms of presentation quality, the paper is clearly written, the proposed methods are well explained, and the notation is consistent.

Overall, a good paper.

Minor comment:
page 3, footnote: "When faced with a non-symmetric matrix, one can resort to the Arnoldi algorithm.": I was wondering if the authors have tried that? I think that the Arnoldi algorithm for non-symmetric matrices are significantly less stable than their Lanczos counterparts for symmetric matrices.

---

> ### Author Response · Authors · 2018-11-20
> **Response**
>
> Thanks for the comments! We have not tried Arnoldi algorithm since we only deal with undirected graphs in the current applications which have symmetric graph Laplacians. Unlike Lanczos algorithm which has error bounds and monotonic convergence properties, Arnoldi algorithm is not well understood since eigenvalues of non-symmetric matrix may be complex and/or badly conditioned. Nonetheless, efficient implementation of Arnoldi algorithm exists. We will explore it in the future.

---

### Author Response · Authors · 2018-11-20
**Common Response**

We thank all the reviewers for the careful reading and the constructive comments. During the rebuttal period, we extended our current model by adding spectral filters for multiple layers, whereas only the first layer contains spectral filters in the submitted version. We show the average results over 3 runs with different random initializations on QM8 as below. Note that experiments of our AdaLanczosNet are still ongoing. We will update this in the later version of our paper.

----------------------------------------------------------------
Methods        | Validation MAE |    Test MAE     |
----------------------------------------------------------------
GCN-FP          | 15.06 +- 0.04      | 14.80 +- 0.09  |
----------------------------------------------------------------
GGNN            | 12.94 +- 0.05      | 12.67 +- 0.22  |
----------------------------------------------------------------
DCNN             | 10.14 +- 0.05      | 9.97 +- 0.09   |
----------------------------------------------------------------
ChebyNet      | 10.24 +- 0.06       | 10.07 +- 0.09 |
----------------------------------------------------------------
GCN                | 11.68 +- 0.09      |11.41 +- 0.10  |
----------------------------------------------------------------
MPNN            | 11.16 +- 0.13       | 11.08 +- 0.11 |
----------------------------------------------------------------
GraphSAGE   | 13.19 +- 0.04       | 12.95 +- 0.11 |
----------------------------------------------------------------
GAT                 | 11.39 +- 0.09       | 11.02 +- 0.06 |
----------------------------------------------------------------
LanczosNet   | 9.65 +- 0.19         | 9.58 +- 0.14   |
----------------------------------------------------------------

---

### Meta-Review · Area_Chair1 · 2018-12-14
**Good paper, recommend for acceptance.**

**Confidence:** 5
**Recommendation:** Accept (Poster)

**Metareview:**

The reviewers unanimously agreed that the paper was a significant advance in the field of machine learning on graph-structured inputs. They commented particularly on the quality of the research idea, and its depth of development. The results shared by the researchers are compelling, and they also report optimal hyperparameters, a welcome practice when describing experiments and results.

A small drawback the reviewers highlighted is the breadth of the content in the paper, which gave the impression of a slight lack of focus. Overall, the paper is a clear advance, and I recommend it for acceptance.